



The Cryosphere

# Reconstruction of annual accumulation rate on firn, synchronising H₂O₂ concentration data with an estimated temperature record

**Jandyr M. Travassos[1], Saulo S. Martins[2], Mariusz Potocki[4,5], and Jefferson C. Simões[3,4]**

[1]Graduate Program in Geophysics, Universidade Federal do Pará (UFPA), Rua Augusto Corrêa n, 01, Belém, Pará, Brazil
[2]Graduate Program in Geology, Universidade Federal Rural do Rio de Janeiro (UFRRJ), 465 km 7, Seropédica, Brazil
[3]Centro Polar e Climático, Universidade Federal do Rio Grande do Sul (UFRGS), Porto Alegre, Brazil
[4]Climate Change Institute, University of Maine, Orono, ME 04469, USA
[5]School of Earth and Climate Sciences, University of Maine, Orono, ME 04469, USA

**Correspondence:** Saulo S. Martins (ssmsaulo@gmail.com)

**Abstract.** This work deals with reconstructing firn layer thicknesses at the deposition time from the firn's observed thickness in ice cores, thus reconstructing the annual accumulation, yielding a timescale and an ice-core chronology. We employed a dynamic time warping algorithm to find an optimal, non-linear alignment between an H₂O₂ concentration data series from 98 m worth of ice cores of a borehole on the central ice divide of the Detroit Plateau, the Antarctic Peninsula, and an estimated local temperature time series. The viability and the physical reliability of the procedure are rooted in the robustness of the seasonal marker H₂O₂ in a high-accumulation context, which brought the entire borehole to within the operational life span of four Antarctic stations around the Antarctic Peninsula. The process was heavily based on numerical optimisation, producing a mathematically sound match between the two series to estimate the annual layering efficiently on the entire data section at once, being disposition-free. The results herein confirm a high annual accumulation rate of $a_N = 2.8$ m w.e./yr TS1, which is of the same order of magnitude as and highly correlated with that of the Bruce Plateau CE1 and twice as large as that of the Gomez Plateau, 300 and 1200 km further south, respectively.

## 1 Introduction

Ice cores provide a continuous record of climatic and environmental data series based on ice's physical and chemical properties, reflecting past atmospheric composition and climatic variability, (e.g. Masson-Delmotte et al., 2006). Snow deposited on the ice surface is gradually compressed into firn and ice, having the ability to preserve a very reliable climate record with a low risk of missing years, provided that the accumulation rate is sufficiently high. A vital issue in the palaeoclimatic reconstruction is dating the stratigraphic sequence through different techniques, including 1-D to 3-D flow models (Nye, 1952; Dansgaard and Johnsen, 1969; Gillet-Chaulet et al., 2012; Passalacqua et al., 2016); the counting cycles of seasonally varying quantities; reference horizons, most commonly layers of high concentrations of sulfuric acid related to volcanic events (Vinther et al., 2006); and layer identification through peaks of radioactive isotopes (Vinther et al., 2006; Cuffey and Paterson, 2010). Often those techniques are combined, e.g. incorporating stable water isotope $\delta^{18}O_{atm}$ into an ice flow model (Capron et al., 2010).

Hydrogen peroxide H₂O₂ is produced by photochemical reactions in natural waters exposed to solar irradiation, surficial and atmospheric. It is the most stable of the reactive oxygen species created in the atmosphere through a chemical reaction requiring ultraviolet light. A kinetic model has explained that 76.7 % of the variation in H₂O₂ concentrations is due to solar irradiance and temperature variation only (Sigg and Neftel, 1988). In particular that production in Antarctica has a pronounced regular seasonality resulting from cycles of complete darkness in midwinter to 24 h daylight in midsummer. This gives a phenomenological basis for quasi-sinusoidal variability in H₂O₂ atmospheric concentration with maxima occurring during the sunlit summer (Steig

et al., 2005; Frey et al., 2006). The $H_2O_2$ is an exceptionally robust marker for ice cores at high-accumulation sites in Antarctica where post-depositional losses are minimised, resulting in excellent preservation of the records, with summer-to-winter ratios of over 5 (Sigg and Neftel, 1988; Hutterli et al., 2003; Frey et al., 2006).

The $H_2O_2$ concentration data come from ice cores extracted from borehole DP-07-1 drilled in December 2007 at the ice divide of the Detroit Plateau (DP), at 64°05′07″ S, 59°38′42″ W, 1930 m above sea level (m a.s.l.). DP-07-1 reveals well-resolved seasonal cycles of $H_2O_2$ concentration data in the context of a very high deposition rate (Potocki et al., 2016). We take advantage of the observed strong seasonality in the $H_2O_2$ record to estimate a core timescale spanning the entire firn horizon. That is done by synchronising the concentration data to an estimated temperature time series at the borehole location.

The maxima of $H_2O_2$ production and surficial atmospheric temperature occur during the sunlit months of the austral summer, allowing us to seek a correlation between their respective maxima. They do not necessarily coincide, but they both occur during summertime; the time difference between them is a fraction of a year. A temperature record at the borehole location on the DP may be estimated by interpolating the historical temperature recordings from six Antarctic stations not too far from the DP: Bellingshausen, Esperanza, Faraday–Vernadsky, Marambio, O'Higgins and Rothera, which have almost continuous meteorological observations from the late 1950s. We have discarded Bellingshausen and O'Higgins, the first for being heavily biased by maritime conditions. The second is a relatively short record with a sizable gap in it, leaving us with four stations forming the vertices of a polygon having the DP within its perimeter. Only Marambio lies on the eastern side, which may imply some unknown bias towards the western temperature regime of the Antarctic Peninsula. Figure 1 shows the locations of the Antarctic stations on an outline of the northern Antarctica Peninsula.

The synchronisation of the concentration data to a temperature series is warranted here due to the local accumulation rate, high enough to bring the entire firn horizon deposition period within reach of the four stations' operational span. Both data series independently follow the same seasonal variation and the passing of the years, albeit in their particular manners; the $H_2O_2$ concentration displays a frequency scaling with depth, a result from the accumulated vertical strain, whereas the temperature series has a uniform frequency behaviour. The frequency scaling reflects the gradual thinning of the annual firn layers, which manifests itself as a frequency chirp in the $H_2O_2$ concentration series.

We have allowed for the frequency scaling of the peroxide concentration series concerning the uniform frequency temperature content by resorting to dynamic time warping (DTW). DTW is a fast and efficient algorithm for finding an optimal alignment between two sequences through a non-

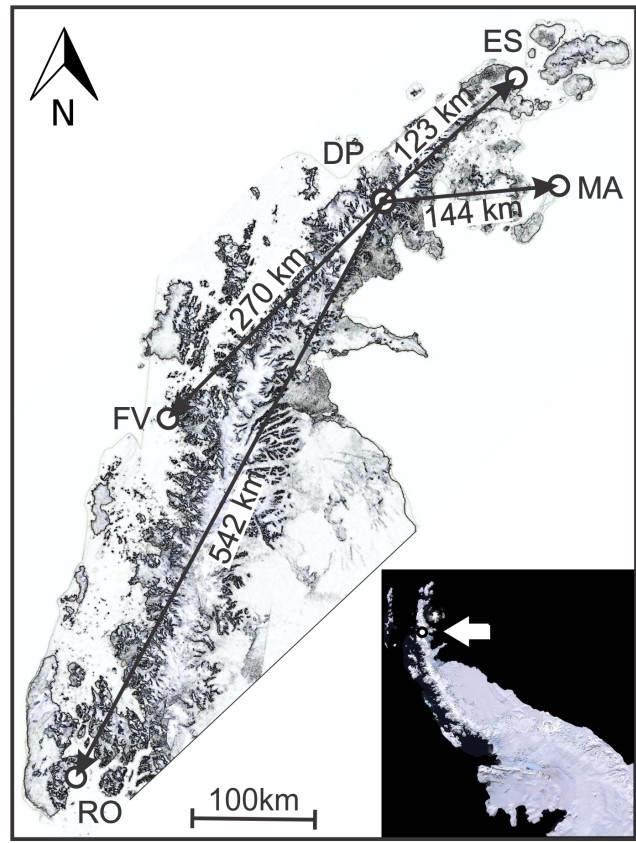

**Figure 1.** The four Antarctic Stations, Esperanza (ES), Marambio (MA), Faraday–Vernadsky (FV) and Rothera (RO), and the borehole at the DP, on the northern Antarctica Peninsula. The white arrow in the lower right corner inset shows the location of the DP on the peninsula. Both maps were modified from a pan-sharpened scene of the Landsat Image Mosaic of Antarctica (LIMA) by USGS (https://lima.usgs.gov/, last access: February 2020 TS2).

linear warping of one onto the other along the time–depth axis (Rabiner et al., 1978; Sakoe and Chiba, 1978). We have worked with standardised versions of the peroxide and temperature series, using the distance between them as a measure for their resemblance (Rabiner et al., 1978; Sakoe and Chiba, 1978). Once this is optimally found, the peroxide series becomes warped onto the temperature series, allowing for the observed frequency scaling.

Notwithstanding DTW being associated with speech recognition (Rabiner et al., 1978; Sakoe and Chiba, 1978; Gilbert et al., 2010), it has proved to be useful in several other applications. These encompass handwriting recognition (Jayadevan et al., 2009), image and shape matching (Wang and Gasser, 1997; Latecki et al., 2007), analysis and classification of the land cover of remotely sensed images (Verbesselt et al., 2010; Xue et al., 2014), gene expression and protein structure (Criel and Tsiporkova, 2005; Legrand et al., 2008), and even brain activity (Chaovalitwongse and Pardalos, 2008). Speech recognition has been used to detect

layers in deep Greenland ice cores, using a hidden Markov model (Winstrup et al., 2010).

This work shows that DTW is also particularly fit for compensating for the peroxide frequency scaling with depth, re-aligning it to a temperature data time series and, at the same time, quantifying their dissimilarities. We have used the constant spectral content of the temperature data series as a reference in the pairing transformation through mathematical optimisation, thus yielding an estimate of a relation of depth to time without human intervention. Moreover, the procedure has also confirmed a very high deposition rate for the entire firn horizon at the DP.

## 2   The data sets

We deal with two independent data sets, a $H_2O_2$ concentration from the 133 m deep borehole and a temperature time series estimated at the DP. We have also collected a record of the stable water isotope deuterium, which was not used due to its poor seasonal variability (Potocki et al., 2016). The borehole yielded intact ice cores down to $z = 109.3 \pm 0.5$ m, from where brittle ice began. The borehole temperature was fairly stable at $-14.2 \pm 0.1\,°C$ at a depth of 10 m. Depths in the borehole are measured with the origin at the surface and the vertical $z$ axis pointing downwards.

The temperature time series at the borehole location was estimated through an interpolation procedure on a data set of continuous temperature readings since 1 January 1970, at four Antarctic stations on the Antarctic Peninsula. We will show below that the entire firn layer was accumulated in a shorter period than the $> 45$ years of the estimated temperature time series.

### 2.1   The $H_2O_2$ concentration data

The $H_2O_2$ concentration data were retrieved from the first 98 m of ice cores with high-resolution sampling, with an average of 36 samples/year. It is a robust seasonal signal, well preserved for the entire depth range of ice cores (Potocki et al., 2016). As for other ice cores at high-accumulation sites across West Antarctica, it is possible to establish a timescale for the core through straightforward counting of the annual cycles (Sigg and Neftel, 1988; Frey et al., 2006).

The $H_2O_2$ concentration record, $C(z)$, has considerable noise content throughout, which has to be minimised, making its seasonal signal conspicuous. We produce a smooth data series $\mathcal{C}(z)$ by robust fitting on $C(z)$ through a loess nonparametric method (Cleveland and Grosse, 1991). Fig. 2 shows both $C(z)$ and $\mathcal{C}(z)$ in micromolar (µM) concentration. It is easy to see the seasonal signal in $\mathcal{C}(z)$ as well as the effect of the accumulated vertical strain with depth on the annual firn layers. The latter manifests itself as a gradual thinning of the annual firn layers.

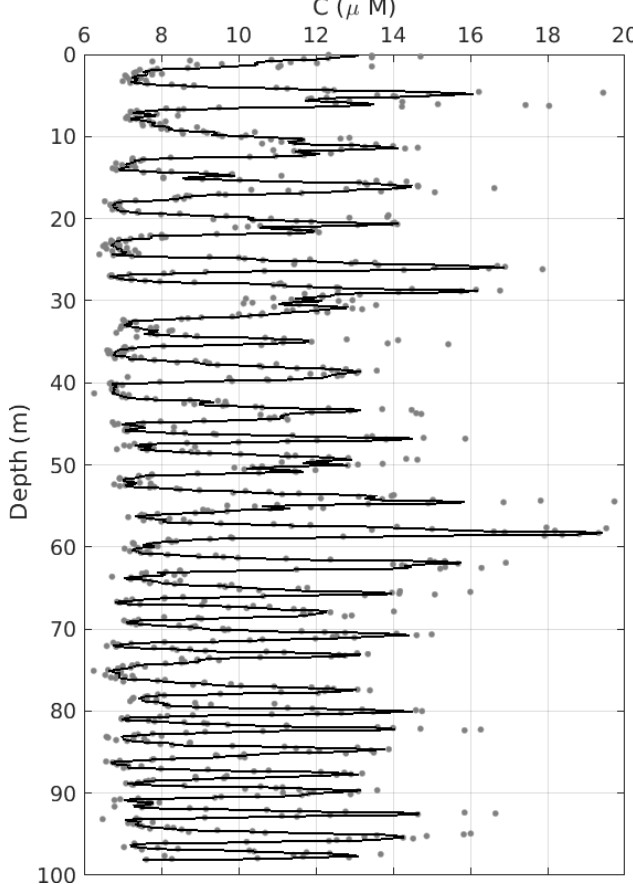

**Figure 2.** The grey dots are the raw data $C(z)$ and the solid line is their smoothed version $\mathcal{C}(z)$, both expressed in micromolar (µM) concentration. For the sake of visualisation, we have omitted just two data points with concentrations of $C(z) > 20\,µM$ at depths approximately equal to 5 m.

Notwithstanding some residual noise left on $\mathcal{C}(z)$, straightforward counting of its peaks and troughs suggests the first 98 m was probably accumulated in its entirety from the beginning of the 1980s. Direct division of the total depth span by the number of peaks indicates a very high deposition rate at the DP, which we will address below.

### 2.2   Estimating a temperature time series at the Detroit Plateau

The four stations shown in Fig. 1 have distinct sampling frequencies of temperature, varying from 1 to 8 readings/day. We set the beginning of the estimated temperature record to 1 January 1970; from this date onward, all four stations have continuous temperature readings. The end of the record is set to 29 December 2010, 3 years after the core was drilled at the DP. These limits yield a period wide enough to encompass the entire deposition period of the firn horizon safely.

We interpolated the daily temperature time series from the four stations shown in Fig. 1 through Delaunay triangulation,

having the DP borehole sea-level projection inside the convex hull formed by the station set. That is a linear interpolation weighted by the inverse of the horizontal distance between a given station and the borehole projection. It is noteworthy that all stations but Marambio are located on the occidental part of the peninsula; nevertheless this station shares the most significant weights with Esperanza. Some bias towards the western climate regime somewhat compensated for by Marambio is thus expected, which is a fact we have to live with anyway.

Only the maximum daily temperature reading at each station was used in the interpolation process. The sea-level-interpolated time series at the DP, $T(t)$, is further corrected to the height of the DP at 1930 m a.s.l., with a lapse rate in temperature with altitude of $-0.55\,°\text{C}/100\,\text{m}$ (Rolland, 2003). Even taking care to obtain the best temperature estimates from the interpolation process at the DP, the accuracy of a particular temperature estimate is not crucial to our results. We use only the location in time of given summertime peak temperatures for synchronisation.

We alleviated aliasing due to the temperature sampling by applying a 2 d low-pass filter to $T(t)$, a series with 14 973 data points, far more than the 985 data points of $C(z)$. We made the number of data points in $\mathcal{T}(t)$ similar to those in $C(z)$ by decimating the former by 15×. Again we avoided aliasing and conspicuously reduced noise in $\mathcal{T}(t)$ by low-pass filtering the decimated data series, using an eighth-order Chebyshev filter. We compensated for the amplitude losses incurred throughout the conditioning process by a constant multiplicative gain, bringing the amplitudes of the filtered temperature time series somewhat back to the original levels of the unfiltered $T(t)$. The multiplicative factor is estimated in successive time windows as the quotient of the envelope of the original $T(t)$ by an envelope of the non-gained version of $\mathcal{T}(t)$. From now on $\mathcal{T}(t)$ will refer to the accumulated temperature time series.

Figure 3 shows the decimated $T(t)$ and $\mathcal{T}(t)$, spanning 41 years. The time series $T(t)$ is quite noisy as one would have expected it to be, but the $\mathcal{T}(t)$ series proves to be a powerful depiction of the annual summer–winter cycles. It has a smaller amplitude than that of $T(t)$, which is hardly an issue here as we are not looking for individual temperature figures but rather a reliable counter for the passing of the years.

## 3 Results

### 3.1 Warping H$_2$O$_2$ concentration data onto the temperature series

Figures 2 and 3 conspicuously show that the $C(z)$ and $\mathcal{T}(t)$ data series record the passing of the years through their annual cycles of peaks and troughs, summer to winter, respectively. Nevertheless the two data series record the annual cycles in distinct manners, the former against depth and the latter against time, their similar shapes suggesting we could employ a simple mapping procedure from depth to year of deposition to a standard variable related to time.

There are two issues to consider here: (i) $C(z)$ and $\mathcal{T}(t)$ have their respective zeniths in a given summer on different dates, as they are distinct phenomena, and (ii) the shapes of the two data series conspicuously differ from each other in terms of their spectral characteristics as is quickly seen comparing Figs. 2 and 3. The first issue is efficiently dealt with as peaks will differ from each other within a fraction of a given summertime, a noise source one just needs to be aware of. The second point is more involved as $\mathcal{T}(t)$ is a function of time with a nearly constant frequency content throughout, whereas $C(z)$ has a frequency scaling with depth, a chirp behaviour easily seen in Fig. 2. The latter results from the gradual thinning of the firn layers due to the weight of the overburden.

The two data series $C(z)$ and $\mathcal{T}(t)$ are not directly comparable, being functions of depth and time. We can make them comparable though by using a standardising mapping procedure:

$$\begin{aligned}
C_i &\longmapsto \widehat{C}_i = \frac{1}{\sigma(C_i)}\left(C_i - \overline{C}\right)\\
\mathcal{T}_i &\longmapsto \widehat{\mathcal{T}}_i = \frac{1}{\sigma(\mathcal{T}_i)}\left(\mathcal{T}_i - \overline{\mathcal{T}}\right),
\end{aligned} \tag{1}$$

where $C_i \equiv C(z)$ and $\mathcal{T}_i \equiv \mathcal{T}(t)$ CE2, with $i = 1, \ldots, N$. $\overline{C}$ and $\overline{\mathcal{T}}$ are averages, and $\sigma(C_i)$ and $\sigma(\mathcal{T}_i)$ are standard deviations. The two standardised series, $\widehat{C}$ and $\widehat{\mathcal{T}}$, have the same number of data points and have zero mean with unit standard deviation. The standardisation process minimises eventual $y$-axis discrepancies between the two series, decreasing the possibility of misalignment by the DTW algorithm. The mapping (1) TS3 is invertible, allowing a return to the original values whenever needed.

We warp the series $\widehat{C}$, call it the sample, onto the reference series, $\widehat{\mathcal{T}}$, allowing for layer thinning with depth in the sample. In applying DTW we construct a warp path $W = (w_1, w_2, \ldots, w_K)$ between sample and reference, where each path element $w_k$ is linked to the two series indexes $(i, i')$, for the $N$ elements in $\widehat{C}$ and $\widehat{\mathcal{T}}$, respectively. The warp path length $W$ is bounded to $N \leq K \leq 2N - 1$ and subject to the criteria below.

- *Boundary conditions.* The warp path starts and ends at the first and the last elements of the two sequences, $w_1 = (1, 1)$ and $w_K = (N, N)$, all elements considered.

- *Continuity.* The warping procedure preserves the ordering of the two aligned sequences:

$$w_k(i, i') \to w_{k+1}(\widehat{i}, \widehat{i'}) \Rightarrow i \leq \widehat{i} \leq (i + 1)$$
$$\text{and } i' \leq \widehat{i'} \leq (i' + 1).$$

- *Monotonicity.* The elements of $W$ are monotonically spaced in the independent variable, thus preventing big

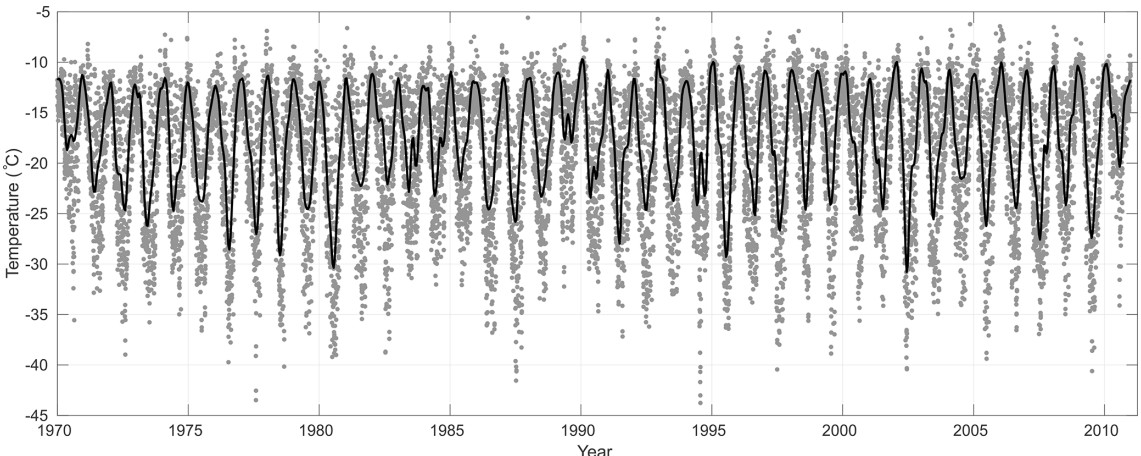

**Figure 3.** The grey dots are the interpolated and decimated temperature time series $T(t)$. The solid curve is its smoothed and gained counterpart $\mathcal{T}(t)$.

jumps:

$$w_k(i, i') \to w_{k+1}(\widehat{i}, \widehat{i'}) \Rightarrow (i - \widehat{i}) \geq 0$$
$$\text{and } (i' - \widehat{i'}) \geq 0.$$

The process of warping the sample onto the reference series is carried out by seeking the path $W$, which yields the minimum distance:

$$D_W = \frac{1}{2N} \min \left\{ \sum_{k=1}^{K} \mathrm{d}(w_k, w_{k+1}) \right\}, \quad (2)$$

where $\mathrm{d}(w_k, w_{k+1})$ is the distance between two contiguous elements. $D_W$ should attain its minimum when the sample is correctly warped onto the reference signal (Sakoe and Chiba, 1978). We do the DTW through an algorithm using a correlation optimised warping, or COW, which aligns the sample onto the reference by piecewise linear stretching and compression of the warping segments with variable lengths $l$ (Nielsen et al., 1998; Pravdova et al., 2002; Tomasi et al., 2004). An integer slack parameter limits the range of possible segments $l$, initially set to unity, $\mathfrak{s} = 1$. The reconstructed sample is obtained by retaining only the highest values obtained for the cumulative correlation coefficient:

$$\xi(\widehat{\mathcal{T}}, \widehat{\mathcal{C}}) = \frac{\sum_l \left(\widehat{\mathcal{T}_{i'}} - \overline{\widehat{\mathcal{T}}}\right)\left(\widehat{\mathcal{C}_i} - \overline{\widehat{\mathcal{C}}}\right)}{(M-1)\,\sigma\left(\widehat{\mathcal{T}_{i'}}\right)\sigma\left(\widehat{\mathcal{C}_i}\right)}, \quad (3)$$

where the summation is performed for each segment $l$ with $M$ points, $\overline{\widehat{\mathcal{T}}}$ and $\overline{\widehat{\mathcal{C}}}$ are averages, and $\sigma\left(\widehat{\mathcal{T}_{i'}}\right)$ and $\sigma\left(\widehat{\mathcal{C}_i}\right)$ are standard deviations. The problem is solved by applying the COW algorithm to all $N/l$ segments through dynamic programming (Nielsen et al., 1998; Pravdova et al., 2002; Tomasi et al., 2004). A complete description of the DTW and COW algorithms is well beyond the scope of this work; the reader is kindly referred to the literature cited herein.

The analysis proceeds as follows: begin the process of warping $\widehat{\mathcal{C}}$ onto $\widehat{\mathcal{T}}$ with the two series aligned at their respective beginnings, the borehole bottom and 1 January 1970, respectively. Warp $\widehat{\mathcal{C}}$ and retain the value of the total distance $D_W$. Discard the year 1970 on $\widehat{\mathcal{T}}$, which now begins on 1 January 1971, and repeat the warping procedure with the entire $\widehat{\mathcal{C}}$ record; retain the new value for the total distance $D_W$. Continue moving forward to the beginning of the $\widehat{T}$ record in 1-year steps, storing the values of $D_w$ estimated at each iteration. Continue this process of advancing the beginning of the $\widehat{\mathcal{T}}$ in 1-year steps, monitoring the evolution of the estimated values of $D_w$.

We observed a decreasing trend in the estimates of $D_w$ retained at each round of warping described above, which reached a conspicuous minimum with the beginning of the $\widehat{\mathcal{T}}$ series aligned on 1 January 1980. Further 1-year steps on the starting date of the temperature ensured an increasing trend with a faster pace. We stopped the 1-year-step warping process on the increasing branch of $D_w$ 5 years after reaching its minimum. Figure 4 shows both the original and the warped versions of series $\widehat{C}$, with the borehole bottom, aligned with $\widehat{T}$ on 1 January 1980. The figure also shows the behaviour of $D_w$ for the entire year span we have considered in our calculations.

Once $\widehat{T}$ is warped onto $\widehat{C}$, one can easily perform an inverse mapping to the original depths and time, with $i = 1, \ldots, N \longmapsto (t; z)$. With that, depths may be mapped onto time, directly yielding a borehole timescale, $z = z(t)$. That is shown in the lower panel of Fig. 4 where $\widehat{C}$ is plotted against deposition time in years. The conspicuous minimum on $D_w$ suggests a quantitative error estimate of $\lesssim 1$ year, significantly greater than any eventual difference between the time of occurrence of the peaks in $\widehat{C}$ and $\widehat{T}$ within a given year.

https://doi.org/10.5194/tc-15-1-2021

The Cryosphere, 15, 1–11, 2021

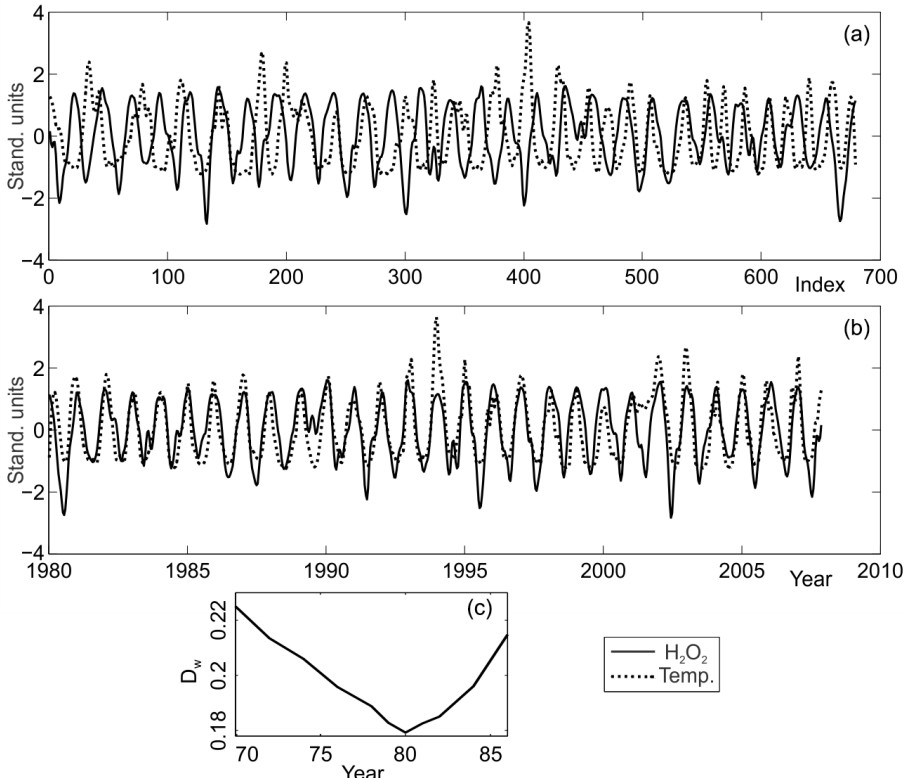

**Figure 4.** Panel **(a)** shows the unwrapped $\widehat{C}$ and $\widehat{T}$ series in standardised ordinates, with the index $i = 0$ corresponding to the mouth of the core. Panel **(b)** shows the two warped series with their abscissas $i$ mapped back to time, expressed in years beginning on 1 January 1980: $\mathcal{T}, \mathcal{C}(t)$. In both panels, $\widehat{C}$ is shown as a dotted curve and $\widehat{T}$ is shown as a solid curve, with ordinates in arbitrary units. Panel **(c)** shows the behaviour of distance $D_w$ for the year we have performed the wrappings.

### 3.2 Estimating a borehole timescale and accumulation rate

A simple model of an ice sheet flow considers that as a year's snowfall moves downwards relatively to the surface during its burial process by subsequent deposition undergoing viscoplastic deformation, it becomes progressively thinner, extending laterally due to ice incompressibility. An increase in density ensues with depth as the snow slowly compacts itself into firn and from that into ice. One way to simplify the process is to express all lengths in water-equivalent units (m w.e.), thus allowing one to disregard the compaction of snow *before* the complete transformation to ice. Accordingly we present depths as $z_{\mathrm{m\,w.e.}} = \frac{z\,\rho(z)}{\rho_w}$, where $\rho(z)$ and $\rho_w$ are the ice cores' density estimates and the density of pure water, respectively.

We use the measured $\rho(z)$ from the ice cores to estimate an empirical model of firn densification, which assumes the density change with depth is proportional to the deviation relative to the density of pure glacier ice $\rho_{\mathrm{ice}} = 0.91 \, \mathrm{g/cm}^3$ (Cuffey and Paterson, 2010). The model may have two (Herron and Langway, 1980) or even three (Ligtenberg et al., 2011) distinct firn densification stages, spanning from the surface to the zone of pore close-off. The adopted model has one densification stage from the surface down to the last available density estimate at $z_{\rho(\mathrm{max})} = 64.5\,\mathrm{m}$: $\rho_z = 0.339\, z^{0.1853}$, with $R^2 = 0.97$ (Travassos et al., 2018). The density measurements beyond $z_{\rho(\mathrm{max})}$ were accidentally lost; so we impose the density of glacier ice onto the core bottom, $\rho(109\,\mathrm{m}) = \rho_{\mathrm{ice}}$, bridging the data gap with a straight line linking the imposed value to the last-measured density. This extrapolation will result in some inaccuracies, but as at $z_{\rho(\mathrm{max})} = 64.5\,\mathrm{m}$ the power law has already reached its slowest increase rate with depth, it may be reasonable to assume they are relatively small. On the other hand, that allows for the transformation of length dimensions to metres water equivalent (m w.e.) for the entire borehole. We will refer to this issue below whenever appropriate.

In the simplest model for an ice sheet flow, the total vertical strain of any layer is equal to the total vertical strain of the ice beneath it:

$$\frac{\lambda(z)}{\lambda_0} = \frac{(1-z)}{h}, \tag{4}$$

where $\lambda(z)$ is a layer of thickness at a depth $z$, which had accumulated as an annual layer of thickness $\lambda_0$, and $h$ is the total ice thickness, from the surface to the bedrock. The model considers a steady-state viscoplastic deformation with depth

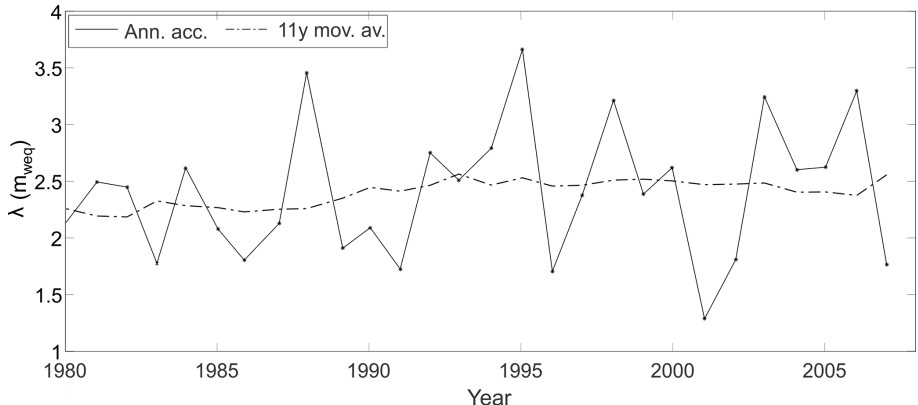

**Figure 5.** The solid line shows the annual accumulation rate estimates at the DP and the dash-dotted line gives their 11-year moving average. Use the right ordinate for the annual accumulation rate and the left ordinate for borehole depths.

at the centre of an ice sheet, as the annual layers are buried by subsequent deposition. From now on, all length dimensions are in metres water equivalent (m w.e.), unless explicitly said otherwise. As the ice sheet is steady, we assume that accumulation and vertical thinning are constant in time and that a layer thickness does not vary horizontally. If those assumptions hold, the distance an ice particle moves downwards during 1 year CE3 must be equal to the thickness of one annual layer $\lambda(z)$ TS4.

As the older ice is closer to the bedrock, it is more convenient to express the vertical position of an ice particle concerning the rock bed interface using a new vertical axis, $Z = h - z$. The new coordinate frame runs in the opposite direction to the one we have been using so far, with $z > 0$ pointing downwards. Assuming a steady state, the distance an ice particle moves downwards in 1 year, or, for that matter, the vertical particle velocity $v(Z)$, is a linear function of $Z$, and therefore, the thinning rate $\frac{dv}{dZ}$ is constant. The velocity at the surface equals the accumulation rate $v(h) = -a$, and at the bed $v(0) = 0$, the velocity being negative in the new reference frame as it points downwards:

$$v(Z) = -a\frac{Z}{h}. \tag{5}$$

The relation between a given depth $Z$ to the age of the ice is provided by

$$t = \int_h^Z v^{-1} dZ \longrightarrow Z = h \exp\left(-\frac{a}{h}t\right), \tag{6}$$

known as Nye's timescale (Nye, 1952, 1963; Cuffey and Paterson, 2010). The relation depicted in Eq. (6) provides the simplest model for describing how a layer of thickness $\lambda_0$ deposited at the surface thins to $\lambda(Z)$ when it is at a distance $Z$ from the bedrock. Notwithstanding its simplicity, the Nye model still provides good estimates at shallow depths, close to the ones from more complex models, such as the

Dansgaard–Johnsen model (Dansgaard and Johnsen, 1969; Cuffey and Paterson, 2010).

The warping of $\widehat{C}$ onto $\widehat{T}$ estimates the deposition thickness $\lambda_0$ from its observed thickness $\lambda(z)$, therefore reconstructing the accumulation as well as yielding a timescale $z(t)$ spanning the entire borehole. The accumulation over the period 1980–2008, as revealed by the warped thicknesses $\lambda_0$, shows wider oscillations towards later years. An 11-year moving average of accumulation shows a fairly stable regime for the period 1980–2008 of $\overline{a}_{11y} \cong 2.5$ m w.e./yr. The small relative increase in accumulation from 1980–1990 to 1990–2008 seen in Fig. 5 is affected by the estimated densities deeper than 64.5 m used to transform depths. Moreover, the statistical significance of an 11-year moving average within a 28-year period is limited; we use it to compare with literature results, where the solar cycle period is often used.

We apply an exponential regression to the warped data to produce estimates for the two constants $\left(h, \frac{a}{h}\right)$ in the relation of Eq. (6). As the available data are confined to the firn layer, an estimate for the total thickness $h$ is obviously beyond our reach.

Nevertheless, as the annual accumulation rate is assumed uniform, we can obtain an estimate for the 27 years before the coring activity of $a_N = 2.82$ m w.e./yr. Peak counting in Fig. 2 yields an estimated accumulation of $a_c = 2.5$ m w.e./yr, equal CE4 to a figure reported elsewhere (Potocki et al., 2016). The two accumulation rate estimates, $a_N$ and $\overline{a}_{11y}$ TS5, differ by $\sim 10\,\%$, demonstrating reasonably compatibility considering the assumptions leading to the relation of Eq. (6) and providing a weak check on our numerical procedure.

It is worthwhile ending this section by comparing our estimated annual accumulation variability with data from the three ice cores listed in Table 1, all south of the DP in the Antarctic Peninsula. Figure 6 shows that the accumulation rates at the DP and Bruce Plateau are compatible throughout, an indication that both sites may have been subject to simi-

**https://doi.org/10.5194/tc-15-1-2021** **The Cryosphere, 15, 1–11, 2021**

**Table 1.** Location of third-party ice cores sites on the Antarctic Peninsula with their distances to the DP ice core. $z_{max}$ is the maximum depth, and the period $\Delta T$, in years, is shown in parentheses.

| Name | Latitude | Longitude | Elevation [m] | $z_{max}$ [m] ($\Delta T$) | Distance [km] | Reference |
|---|---|---|---|---|---|---|
| Bruce Plateau | −66.0 | −64.1 | 1976 | 448 (1750–2009) | 302 | Goodwin et al. (2016) |
| Dyer Plateau | −70.7 | −64.9 | 2002 | 190 (1504–1990) | 767 | Thompson et al. (1994) |
| Gomez Plateau | −73.6 | −70.4 | 1400 | 136 (1858–2006) | 1137 | Thomas et al. (2008) |

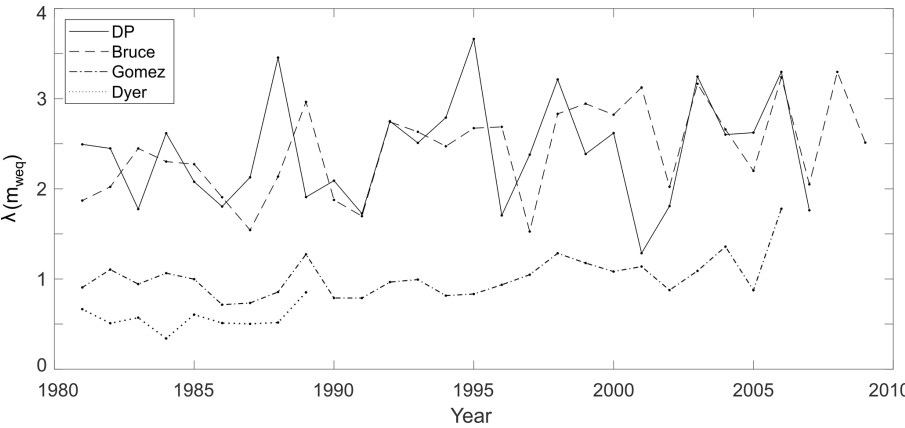

**Figure 6.** Annual snow accumulation in ice cores from the DP (solid line), Bruce Plateau (dashed line), Gomez Plateau (dash-dotted line) and Dyer Plateau (dotted line) for the period 1980–2010.

lar high-accumulation regimes, twice as large as that of the Gomez Plateau. Figure 6 also suggests annual snow accumulation for the period 1980–2010, giving a stable accumulation for all four ice cores. Nevertheless, the period spanned by our data is too short for probing multi-decadal trends; it has been reported that the Antarctic Peninsula has been experiencing an increased rate since 1900 (Thomas et al., 2017). In particular, the Bruce Plateau ice core suggests an increase in snow accumulation during the late twentieth century, increasing at a rate of 0.19 mm w.e./yr since the 1950s (Goodwin et al., 2016).

## 4   Conclusions

Stratigraphic dating of ice cores is rooted in the use of reference horizons and annually resolved data to count annual layers to establish a core chronology. The latter uses outward data, e.g. volcanic events, to measure annual layers. This work has resorted to an independent data set, recorded temperature series, as a time reference to reconstruct a given layer thickness $\lambda_0$ at deposition time from its observed thickness $\lambda(z)$, thus reconstructing the annual accumulation and thereby a timescale or an ice-core chronology $z(t)$.

The adopted non-linear numerical algorithm warped the $H_2O_2$ concentration data from borehole DP-07-1 onto an estimated local temperature record by aligning their respective summertime peaks, an interannual process with a $\simeq 0.5$-year time accuracy. Both the viability and the physical reliability

of the procedure were rooted in the robustness of $H_2O_2$ as a seasonal marker associated with the observed high accumulation rate, which brought the entire borehole to within the operational life span of the Antarctic stations.

The considerable noise content in both series was alleviated through a nonparametric loess filter, which produced clean, smoothed versions of the data series albeit still retaining their complexity, as seen in Figs. 2 and 3. Any time difference between the summertime temperature and peroxide concentration peaks falls necessarily within the interannual process' time accuracy of $\simeq 0.5$ years. The whole process was based on numerical optimisation, producing a mathematically sound match between the two series.

The secular variation in accumulation has revealed a high annual accumulation rate of $a_N = 2.8$ m w.e./yr, with the large variability seen in Fig. 5. The high accumulation rate observed at the DP is of the same order as the one reported for the Bruce Plateau, and they are highly correlated throughout the observational period considered here. The DP regime occurs 1 year earlier than at Bruce in a couple of time sections in Fig. 6, a small but detectable discrepancy, probably related to the distinct dating approaches. The conspicuous correlation of the DP and Bruce is an indication that the northern tip of the Antarctic Peninsula has been under a high-snow-accumulation regime, twice as large as that of the Gomez Plateau further south. The short period reported here is incapable of revealing multi-decadal trends; nevertheless, it is reasonable to suggest the DP may have been experiencing a

similar increase in snow accumulation in the late twentieth century, similar to the one reported at the Bruce Plateau.

The limited time window of the period of our data reveals relatively stable behaviour throughout the 27 years before coring, with an 11-year moving average of the accumulation of $\overline{a}_{11y} \cong 2.5\,\mathrm{m\,w.e./yr}$. A regularity in snow deposition preserved a reliable climate record, minimising post-depositional losses in the concentration of $H_2O_2$. We should expect a relatively short temporal range for firn layer ice cores in the northern Antarctic Peninsula by the same token, turning that region into a valuable climate record ranging through 3 decades before coring. The top DP layer should be now, almost 15 years after drilling, halfway through the firn layer if assuming deposition rate stability.

Mathematical procedures for annual layer counting are notoriously more laborious than manual counting; nevertheless, the latter has no other intrinsic quality but its easiness; quality or effectiveness cannot be technically guaranteed. As is the case of the present work, the former approach is indisputably rigorous, able to efficiently estimate the annual layering on the entire data section and disposition-free. The layer counting applied to our data produced annual accumulation figures that differ from those presented here by up to 40 %, with 17 % on average. All that considered, the choice ultimately remains with the investigator weighing in on an acceptable level of chronological inaccuracy in their work.

Comparison of algorithm results with simple layer counting performed on the smoothed versions of our data set suggests inaccuracies are non-uniform and within ±1 year. Notwithstanding that the algorithm is potentially usefully applied to other data sets where manual counting is more challenging than in the present case, it is not case-specific, and it is not restricted to the dyad peroxide temperature either; it can deal with other kinds of annually laminated data, not necessarily of a related origin, even among different wells. We are convinced there may be many different situations where there is the need to synchronise particular data sets where the procedure shown here may prove helpful.

*Code availability.* We have modified somewhat a COW code from (Tomasi et al., 2004). The original code is available at http://models.kvl.dk/dtw_cow.

*Data availability.* – *Temperature.* The interpolated temperature daily series at the sea-level projection of the borehole DP-07-1 we have used is in the file "daily_temperature.asc" published at https://doi.org/10.6084/m9.figshare.14946177.v1 (Martins and Travassos, 2021) TS6.

– $H_2O_2$. The peroxide concentration data can be requested from Mariusz Potocki TS7 (mariusz.potocki@maine.edu). TS8 CE5

*Author contributions.* JMT worked with the synchronisation of $H_2O_2$, the temperature data series and its accrued accumulation rate and wrote the manuscript. SSM estimated the temperature data series and contributed with additional data processing. MP processed the original $H_2O_2$ data series from the ice cores. JCS worked with all aspects of acquiring the ice cores in the field and contributed to several glaciological aspects of this work. All authors reviewed and agreed on the final manuscript.

*Competing interests.* The authors declare that they have no conflict of interest.

*Acknowledgements.* This work was fully supported by the Brazilian Antarctic Program (PROANTAR) through the CNPq and CAPES. The present work is part of the ice-core programme Climate of Antarctica and South America (CASA) in association with the Climate Change Institute, University of Maine. The authors also acknowledge the Chilean Antarctic Institute (INACH), the Chilean Air Force (FACh), the Brazilian Air Force and the Brazilian Navy.

*Financial support.* This research has been supported by the INCT da Criosfera (CAPES project 88887.136384/2017-00) and PROANTAR–CNPq (project 442755/2018-0). TS9

*Review statement.* This paper was edited by Michiel van den Broeke and reviewed by two anonymous referees.

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

## Remarks from the language copy-editor

## Remarks from the typesetter