# Peer review of "Reconstruction of annual accumulation rate on firn synchronizing $H_2O_2$ concentration data with an estimated temperature record"

_The Cryosphere, 2020_

## Author Comment (AC1)

**MAJOR COMMENTS**

**l. 34: "surficial atmospheric temperature alone as a proxy for the solar irradiation" Why would the temperature be a good proxy for insolation? It is well known that the seasonal cycle in temperature lags that of insolation, especially in regions where horizontal advection is important and when open seas are nearby. Moreover, sea ice cover may influence the seasonality, depending on wind direction and time of year. Please provide more evidence that the used temperature record lines up well with (top-of-atmosphere) insolation, and how discarding the Bellingshausen record makes a difference. Another useful addition might be to select subsets of the four stations to study the dependency of the final result on station selection, and compare what happens if e.g. simple time series of top of atmosphere insolation is selected as a counter of the passing of the years.**

*Answer:*

*The selected text above is an unfortunate flaw in the manuscript. Please accept our apologies for it. We did not use "of the surficial atmospheric temperature alone as a proxy for the solar irradiation" at all in the paper. What we have done was to synchronize the peaks of the peroxide record to the summertime peaks in the temperature record, the latter acting as a time reference. Sure they do not necessarily occur at the same time on a given year, as the local surface temperature does not correlate perfectly with local insolation at a given instant. Nevertheless, the difference between the summertime peaks of the peroxide and the temperature is of the order of or less than 0.5 years. Our work deals with annual estimates therefore that discrepancy is less than the temporal resolution of our data.*

*We have made the following amendments in order to correct the manuscript.*

**Further to the above: what potential role does precipitation seasonality play in influencing the signal? And how sensitive are your results to the filtering applied to both time series?**

*Answer:*

*On this issue, we draw your attention to Figs. 2 and 3. Please note that the filter is very robust face the considerable noise content on both series, any eventual displacement either in depth, Fig. 2, or in time, Fig. 3, will remain within a fraction of a year thus less than the temporal resolution of our data. That considered we found your query should be properly addressed in the text, so we have enlarged the first paragraph of the Conclusions to:*

*"The $H_2O_2$ concentration record, C (z) has a considerable noise content throughout, which has to be minimized making its seasonal signal conspicuous. We produce a smooth data series **C (z)** by robust fitting on C (z) through a loess nonparametric method (Cleveland and Grosse, 1991). That filter is robust enough face of noise to render any eventual displacement in depth of a given concentration peak correspond to a fraction of a year, less than the temporal resolution of our data.*

*We have produced a time scale, an ice-core chronology z(t) for the 133m deep borehole DP-07-1 drilled in Detroit Plateau, Antarctic Peninsula, as a direct result of the process of warping a high resolution $H_2O_2$ concentration data series onto an estimated local temperature time series. The physical reliability of the procedure is rooted both on the*

*robustness of $H_2O_2$ as a seasonal marker and on the local high accumulation rate, which brought the borehole depth span to within the operational life span of the Antarctic stations. The adopted numerical procedure, a non-linear warping of the concentration data onto the estimated temperature record, was performed aligning their respective summertime peaks, thus requiring a conspicuous seasonal signal throughout. Both series have a considerable noise content rendering peak identification a considerable challenge prone to disposition. We alleviated noise through a loess nonparametric filter, which produced clean smoothed versions of the data series albeit still retaining their complexity, as seen in Figures 2 and 3. Any eventual displacement in depth or time of a given concentration peak correspond to a fraction of a year, which is less than the temporal resolution of our data.*

*The application of the non-linear warping algorithm on the smoothed data series allowed for the correction of the thinning with depth of the firn annual layers due to the vertical strain, yielding an estimate of their thicknesses at the time of snow deposition, not considering the snow which was eventually displaced by the wind. The whole process is all based on numerical optimization, yielding the best match between the two series and a secular variation in local precipitation, characterized by the relatively large interannual accumulation rate variability easily seen in  Figure 5"*

**An important outcome of this work is not only the average accumulation rate but also the interannual accumulation variability, which is very large (Fig. 5). To enhance the impact of the paper I would like to see a direct comparison of the annual accumulation time series as obtained from this study and as obtained from simple layer counting, as often done in glaciology.**

*Answer:*

*We do agree that simple layer counting would give similar results to our results. But the algorithm is able to determine the most likely annual layering in an entire data section at once, while being based on purely mathematical objective criteria. Our intention was to demonstrate the effectiveness of our method on a dataset we have collected. The method may prove to be useful in other data where manual counting is more challenging than the present case.*

*We have done the layer counting using the smoothed versions of the series but we don't agree we should present that in our paper. I believe one cannot say that one approach is better than the other, for sure the manual counting is quicker, but you may agree with us that using the algorithm is a safeguard against disposition.*

*But inspired by your observation on the interannual accumulation variability we decided to include a comparison of our data with three other sites: Gomez, Dyer Plateau and Bruce Plateau. You may agree this addition may enhance the impact of the paper. We found it useful, please see below.*

[Figure]

But we do agree your point should be addressed in the text, so we have changed the 2nd paragraph of the Conclusions

*From*
*The adopted numerical procedure, a non--linear warping of the concentration data onto the estimated temperature record, was performed aligning their respective summertime peaks, thus requiring a conspicuous seasonal signal throughout. Both series have a considerable noise content rendering peak identification a considerable challenge prone to disposition. We alleviated noise through a loess nonparametric filter, which produced clean smoothed versions of the data series albeit still retaining their complexity, as seen in Figures 2 and 3. Any eventual displacement in depth or time of a given concentration peak correspond to a fraction of a year, which is less than the temporal resolution of our data.*

*To*
*The adopted numerical procedure, a non-linear warping of the concentration data onto the estimated temperature record, was performed aligning their respective summertime peaks. Both series have a considerable noise content which renders peak identification a considerable challenge. We alleviated that noise through a loess nonparametric filter, which produced clean smoothed versions of the data series albeit still retaining their complexity, as seen in Figures 2 and 3.*
*Note that the summertime temperature and peroxide concentration peaks do not necessarily occur at the same time on a given year, as the local surface temperature does not correlate perfectly with local insolation. Nevertheless the expected difference between the summertime peaks of the two series are of the order of 0.5 year, less than the temporal resolution of our data which is one year.*
*Here it is important to say that simple layer counting would give somewhat similar results to the ones presented here, but it is important to note the considerable noise content renders peak identification a considerable challenge, prone to disposition. Manual layer counting on the smoothed versions of the data series gives interannual accumulation figures that may differ from the ones presented here up to 40%, 17% on average. The algorithm is able to determine the most likely annual layering in an entire data section at once, while being based on purely mathematical objective criteria.*

**Minor and textual comments**

*We have done all the suggested modifications. In some we felt we should clarify better or add comments of our own. Those are dealt with below.*

**l. 19: The H2O2 -> Hydrogen peroxide (H2O2)**
*Done*

**l. 19: "surficial and atmospheric" Do you refer to H2O2 or solar radiation? Unclear what you mean here, please reformulate**
*From:*
*"The reported strong seasonality on the production of $H_2O_2$ in Antarctica, allow us to concentrate on the yearly cycles of the surficial atmospheric temperature alone as a proxy for the solar irradiation."*

*To:*
*The maxima of $H_2O_2$ production and of surficial atmospheric temperature occur during the sunlit months of the austral summer, allowing us to seek a correlation between their respective maxima. Obviously they do not necessarily occur at the same time but they do during summertime, the time difference between them being a fraction of a year.*

**l. 26: Can it be briefly explained why the concentration ratios differ by an order of magnitude between atmosphere and snow? What about the diffusion of the signal in the ice core?**
*A:*
*The accumulation at Detroit Plateau is very high minimizing post--depositional losses from degassing, resulting in an excellent preservation of the record. In particular I could not trace the one order of magnitude difference between atmosphere and snow at a particular site in the literature I used; it seems to be a typo. In face of that we changed text*
*From:*
*"The $H_2O_2$ reportedly may reach a summer-to-winter ratio of 5 in high accumulation rate ice cores from atmospheric concentration ratios of ~ 50 (Sigg and Neftel, 1988; Hutterli et al., 2003; Frey et al., 2006). The $H_2O_2$ is a particularly robust marker for ice cores at high accumulation sites in Antarctica (Sigg and Neftel, 1988; Frey et al., 2006)."*

*To:*
*"The $H_2O_2$ is a particularly robust marker for ice cores at high accumulation sites in Antarctica (Sigg and Neftel, 1988; Frey et al., 2006) where post-depositional losses are minimized resulting in excellent preservation of the records, with summer-to-winter ratios in excess of ~ 50 (Sigg and Neftel, 1988; Hutterli et al., 2003; Frey et al., 2006)."*

**l. 29: Plateau Detroit -> Detroit Plateau (throughout, please)**
*Done*

**l. 34: surficial atmospheric temperature alone as a proxy for the solar irradiation -> near-surface (2 m) atmospheric temperature alone as a proxy for the solar irradiation**
*A: We have changed this part of the text.*

**l. 68: "conductivity measurements on ice cores down to 20m had a modal value of 40.4µS/cm" What is the added value of this information?**
*A: None, withdrawn.*

**l. 75: "along the 98m of ice cores" Earlier, ice core length was 133 m, with intact ice**
*Text changed accordingly,*
*from (1st paragraph of Section 2.1)*
*"concentration data has a high-resolution sampling, averaging to 36 samples/year along the 98m of ice cores."*
*To*
*"concentration data  was retrieved from the first $98\unit{m}$ of ice cores with high-resolution sampling, with an average of 36 samples/year."*

**l. 84: "the first 100m" See above.**
*A: Corrected to 98m.*

**l. 94: "We have considered 95 only the maximum daily temperature reading at each station" Why? When was the reading taken at the station with one reading per day?**
*A: We don't know when readings were taken. We just retained one, the maximum daily reading at each station. Obviously if a Station has one reading, we used it.*

**l. 96: ", using a conservative lapse rate for the decreasing of temperature with altitude of −0.55â¦C/100m"  Since you have a good estimate of the annual mean surface temperature at Detroit Plateau (being the 10 m firn temperature, assuming no meltwater refreezing), you can estimate the temperature lapse rate yourself, neglecting the temperature difference between surface and 2 m.**
*A: You are right on this. It is embarrassing we did not simply use that. Our estimated temperature lapse rate is -0.45C/100m. Using this would have implied a rise in the average temperature at 1937m of +1.940C. This difference does not impact our results.*

**l. 155: Begin this line with a small introductory remark, e.g.: "The analysis proceeds as follows: "**
*Done*

**l. 164: Typo "increasing"**
*Done*

**l. 196: Typo "As ice sheet"**
*Done*

**l. 216: Why was an 11-year moving average chosen?**
*A: It was a choice of ours, we have used the period of a solar cycle.*

**l. 222: Although both accumulation estimates are close, they still differ by more than 10%. Is this within the range of expectations?**

*A: We believe so. The two accumulation rate estimates are reasonably compatible considering the assumptions leading to Nye's time scale. This provides a weak check on our numerical procedure.*

**l. 234: Typo "equals"**
*Done*

**l. 238: Add "of" between "Peninsula" and "0.8"**
*Done*

*A: We believe so. The two accumulation rate estimates are reasonably compatible considering the assumptions leading to Nye's time scale. This provides a weak check on our numerical procedure.*

---

## Author Comment (AC2)

This study reports an ice-core dating method, based on a non-linear pairing transformation of $H_2O_2$ concentration data and a time series of estimated temperature, for the chronology of 113m deep borehole from Detroit Plateau at the Antarctic Peninsula. The thinning of annual firn layers is considered in this method. According to the chronology, combining with snow density, snow accumulation rate is determined during 1980-2010.

Ice core dating is a primary prerequisite for recovering climatic and environmental information using ice core records. The dating method presented here is new and important. The manuscript is well organized and well written. The figures are interesting. In my opinion, the manuscript should be accepted after addressing the following comments.

**Main comments**

1. **Despite the importance of the presented dating method, I think it is difficult to be widely used for other ice core dating over Antarctica, because the long-term temperature observations are too sparse. Therefore, its potential applications should be carefully clarified to add the value of this study.**

   *Answer:*

   *We do agree temperature observations may not be widely available over Antarctica, but the algorithm can be used to synchronize distinct datasets, so be used in other contexts, like synchronizing distinct boreholes. Your comment is addressed on the Conclusions in a new last paragraph:*

   *Our goal was to demonstrate the effectiveness of our method on a dataset we have collected on PD. We believe the method may prove to be useful in other data where manual counting is more challenging than the present case. Moreover we draw the attention of the reader to the fact the algorithm showed to be effective in synchronizing distinct datasets, so it can be used in other contexts, like synchronizing distinct boreholes.*

2. **The authors make so many efforts on the chronology, and seem to only obtain the important accumulation rate results, which are easily determined by layer counting. This greatly reduce the scientific value of the present manuscript. So it is necessary to clarify the priority of your method relative to layer counting after a comparison.**

   *Answer:*

   *We do agree that simple layer counting is a great deal easier and that it would give somewhat similar results to our dataset, albeit not being possible to assure one is better than another. We did layer counting on the smoothed series as an internal check for our results.*

   *Purely mathematical procedures for annual layer counting are laborious compared to manual counting, nevertheless the latter has no other intrinsic*

*quality but easiness; quality or effectiveness cannot be technically guaranteed. Mathematical approaches are indisputably rigorous, disposition--free and, in our case, able to estimate efficiently the annual layering on the entire data section at once. It directly produces inter--annual layering and the annual accumulation rate. Notwithstanding being used in a particular dataset it the algorithm is not case—specific and so it can be used with other datasets and sites. It is not even bound to the dyad peroxide--temperature; it can deal with other kinds of annually laminated data, like from distinct wells. Moreover the algorithm may prove to be useful in other data where manual counting is more challenging than the present case.*

*Here is important to note that simple layer counting would give somewhat similar results to the ones presented here, but it is important to note the considerable noise content renders peak identification a considerable challenge, prone to disposition. Manual layer counting on the smoothed versions of the data series gives inter--annual accumulation figures that differ from the ones presented here up to 40%, being 17% on average.*

*Our goal was to demonstrate the effectiveness of our method on a dataset we have collected on PD. We believe the method may prove to be useful in other data where manual counting is more challenging than the present case. Moreover we draw the attention of the reader to the fact the algorithm showed to be effective in synchronizing distinct datasets, so it can be used in other contexts, like synchronizing distinct boreholes.*

*We have added the above to the end of the Conclusions.*

3. **To further add the scientific values, interpretation of cause of the resulting snow accumulation rate changes since 1980 is required. I also would like to see further comparison of this time series with other previously published ice core snow accumulation over the Antarctic Peninsula.**

*Answer:*

*We have added the following paragraph on that in the Conclusions section:*

*"Our results contribute to confirm the Antarctic Peninsula region of high snow accumulation, allowing a rapid sequestration of the seasonal deposition of chemical species, at least the $H_2O_2$. Conversely the high annual accumulation limits the temporal range of ice cores in this region, confining records to the second half of the twentieth century. The ice core accumulation figure may represent mass gain to the local ice sheet, an important component of the total Antarctic mass balance."*

*We have included a comparison of the interannual accumulation variability in our data with three other sites, Gomez, Dyer plateau and Bruce plateau. This discussion comes at the end of Section 3.2, just before the Conclusions. Your suggestion was greatly appreciated. It follows the material added to the manuscript.*

*"It is worthwhile to end this section comparing our estimated annual accumulation variability with data from the three ice cores listed in Table 1, all South of PD in the Antarctic Peninsula. Figure 6 shows that the accumulation rates at PD and Bruce Plateau are compatible throughout, an indication that both sites may have been subject to similar high accumulation regimes, twice as large as Gomez's. The Figure also suggests annual snow accumulation for the period 1980–2010 a stable accumulation for all four ice cores. Nevertheless the time period spanned by our data is too short to probe multi–decadal trends, it is reported that the Antarctic Peninsula has been experiencing an increased rate since 1900 (Thomas et al., 2017). In particular the Bruce plateau ice core suggests an increase in snow accumulation during the late twentieth century, increasing at a rate of 0.19 mm w.e./y since the 1950's (Goodwin et al., 2016)"*

**Table 1.** Location of third party ice cores sites on the Antarctic Peninsula with their distances to PD ice core. $z_{max}$ is the maximum depth and the time span $\Delta T$, in years, is shown between square brackets.

| Name | Latitude | Longitude | Elevation(m) | $z_{max}(m)[\Delta T]$ | Distance(km) | Reference |
|------|----------|-----------|--------------|------------------------|--------------|-----------|
| Bruce plateau | -66.0 | -64.1 | 1976 | 448[1750–2009] | 302 | (Goodwin et al., 2016) |
| Dyer plateau | -70.7 | -64.9 | 2002 | 190[1504–1990] | 767 | (Thompson et al., 1994) |
| Gomez | -73.6 | -70.4 | 1400 | 136[1858–2006] | 1137 | (Thomas et al., 2008) |

[Figure]

*NB: Both the Table and the Figure above are screen snips; they do not reproduce the manuscript's quality.*

4. **This manuscript gives results, but not discuss them.**

   *Answer:*

   *Agreed, it was a bit too descriptive. We have enlarged the Conclusions accordingly to include a discussion on our results.*

Minor comments

**Line 1 Change "peroxide, H₂O₂," to "peroxide (H₂O₂)"**

*Ans: Done*

**Line 11 "e.g. Masson-Delmotte et al. (2006)." should be "(e.g., Masson-Delmotte et al., 2006)"**

*Ans: Done*

**Line 29 Change "Plateau Detroit" to "Detroit Plateau", and check throughout the text.**

*Ans: Done*

**Line 93-97 Please give some discussion on the uncertainty of the interpolation.**

*Ans: We have expanded the paragraph to two, to accommodate a discussion on the interpolation process. We cannot fathom the uncertainty on the temperature estimates but it is accepted that Delaunay triangulation minimizes interpolation errors (Chen, Long, and Jin-chao Xu. "Optimal delaunay triangulations." Journal of Computational Mathematics (2004): 299-308). Moreover the accuracy of a particular temperature estimate is not crucial to our results as we use only the location in time of a given summertime peak temperatures for synchronization, not the temperature values.*

**Line 215-218, The determined snow accumulation time series is only 28 year, and 11-year moving average is statistical significance? Please explain this.**

*Ans: You are right on this. The statistical significance of the 11-year moving average (period of a solar cycle) is rather limited. We have used that for the sake of comparison with other author's results, as it is frequently used. We have also changed to entire paragraph to address your comment.*

**Figure 4, suggest to use full lines and dotted line to discriminate H₂O₂ and temperature more clearly.**

*Ans: We've done that. It did improve the Figure. Thanks.*

**Figure 5, the horizontal ordinate is vague.**

*Ans: We have simplified the Figure 5, leaving only the interannual accumulation and the 11-year moving average at PD. We believe now the function of the horizontal axis became clearer. We have included the accumulation in other sites in a new Figure 6, as for your major comment 3.*